# Bimetallic nickel-molybdenum/tungsten nanoalloys for high-efficiency hydrogen oxidation catalysis in alkaline electrolytes

Yu Duan[1,6], Zi-You Yu[1,6], Li Yang[2,3,6], Li-Rong Zheng[4], Chu-Tian Zhang[1], Xiao-Tu Yang[1], Fei-Yue Gao[1], Xiao-Long Zhang [1], Xingxing Yu[1], Ren Liu[1], Hong-He Ding[5], Chao Gu[1], Xu-Sheng Zheng[5], Lei Shi[1], Jun Jiang [2], Jun-Fa Zhu [5], Min-Rui Gao [1✉] & Shu-Hong Yu [1✉]

Hydroxide exchange membrane fuel cells offer possibility of adopting platinum-group-metal-free catalysts to negotiate sluggish oxygen reduction reaction. Unfortunately, the ultrafast hydrogen oxidation reaction (HOR) on platinum decreases at least two orders of magnitude by switching the electrolytes from acid to base, causing high platinum-group-metal loadings. Here we show that a nickel-molybdenum nanoalloy with tetragonal $MoNi_4$ phase can catalyze the HOR efficiently in alkaline electrolytes. The catalyst exhibits a high apparent exchange current density of 3.41 milliamperes per square centimeter and operates very stable, which is 1.4 times higher than that of state-of-the-art Pt/C catalyst. With this catalyst, we further demonstrate the capability to tolerate carbon monoxide poisoning. Marked HOR activity was also observed on similarly designed $WNi_4$ catalyst. We attribute this remarkable HOR reactivity to an alloy effect that enables optimum adsorption of hydrogen on nickel and hydroxyl on molybdenum (tungsten), which synergistically promotes the Volmer reaction.

[1] Division of Nanomaterials and Chemistry, Hefei National Laboratory for Physical Sciences at the Microscale, Institute of Energy, Hefei Comprehensive National Science Center, CAS Center for Excellence in Nanoscience, Department of Chemistry, Institute of Biomimetic Materials and Chemistry, University of Science and Technology of China, 230026 Hefei, China. [2] Hefei National Laboratory for Physical Sciences at the Microscale, CAS Center for Excellence in Nanoscience, School of Chemistry and Materials Science, University of Science and Technology of China, 230026 Hefei, China. [3] Institutes of Physical Science and Information Technology, Anhui University, Hefei, Anhui 230601, China. [4] Beijing Synchrotron Radiation Facility, Institute of High Energy Physics, Chinese Academy of Sciences, 100049 Beijing, China. [5] National Synchrotron Radiation Laboratory, University of Science and Technology of China, 230029 Hefei, China. [6] These authors contributed equally: Yu Duan, Zi-You Yu, Li Yang. ✉email: mgao@ustc.edu.cn; shyu@ustc.edu.cn

Over the past few years, market penetration by hydrogen fuel cell vehicles has begun owing to the tremendously advanced proton exchange membrane fuel cell (PEMFC) technologies[1]. Nevertheless, considerable market barriers still exist because current PEMFCs rely heavily on platinum (Pt)-based catalysts that drive the sluggish cathode oxygen reduction reaction (ORR) at low pHs, which raises poor cost competitiveness[2]. Alternatively, hydroxide exchange membrane fuel cells (HEMFCs) give critical merits over PEMFCs, which permit the adoption of Pt group metal (PGM)-free catalysts to negotiate the formidable ORR[3–5], leading to substantial cost reduction. Unfortunately, the anode hydrogen oxidation reaction (HOR) activity on PGM catalysts (e.g., Pt, Ir, and Pd) is about two to three orders of magnitude slower in alkali than in acidic electrolytes[6,7]. This consequently causes higher PGM loadings at the anode that could largely offset the reduced cost from the use of PGM-free cathodes[8]. As a result, the development of durable PGM-free catalytic materials with high intrinsic HOR activity in alkali is important to the eventual success of HEMFC technology.

Recent advances in the design of alkaline HOR catalysts and related mechanistic understanding have primarily focused on PGMs and their alloys[9–11]. Numerous PGMs, such as Pt, palladium (Pd), iridium (Ir), ruthenium (Ru), and rhodium (Rh,) have been studied for HOR in alkaline electrolytes[10,12], among which Pt and Ir are particularly active and stable. Moreover, alloying PGMs with other metals can enable performance enhancements resulting from the modified surface structures; typical examples include PtNi[13], PtRu[14], Pt-coated Cu[15], body-centered cubic PdCu[9], and others[16–20]. In the quest to understand why the HOR reactivity in alkaline media is significantly slower than that in acid on PGMs, there has been extensive debate over whether such sluggish HOR kinetics in alkali is determined by hydrogen binding energy (HBE) or OH binding energy (OHBE)/ oxophilicity[8,10,11,13,21–27]. The lack of a conclusive mechanistic relevance has somewhat hampered success in designing better-performing HOR catalysts from PGMs. With regard to cost-effective HEMFC anode, replacement of the PGMs with PGM-free catalysts—for example, nickel (Ni)[28], $Ni_3N$[29], Ni/CeO$_2$[30], Ni/NiO/C[31], NiMo/C[32], CoNiMo[33], Ni/N-doped carbon nanotubes[34], and Cr-decorated Ni[35]—has been intensively proposed. However, the HOR activity and durability of these Ni-based catalysts are ordinary. To our best knowledge, although numerous efforts have been devoted to developing PGM-free catalysts for alkaline HOR since 1960s[36], no catalyst with activity superior to commercial Pt/C has been reported, which severely limits their practical adoption in HEMFCs.

Herein, we report an important development in totally PGM-free HEMFC anode by using bimetallic MoNi$_4$ alloy as a catalyst, which enables the HOR catalysis in alkaline electrolytes highly efficient. The nanostructured MoNi$_4$ catalyst yields a geometric exchange current density of 3.41 mA cm$^{-2}$ towards the HOR, which is 1.4 times higher than that of commercial Pt/C catalyst and compares superior to previously reported PGM-free catalysts measured in alkali. At 50 mV, a geometric kinetic current density of 33.8 mA cm$^{-2}$ is obtained for MoNi$_4$ catalyst, which represents 105- and 2.8-fold increase as compared to the freshly synthesized Ni and commercial Pt/C catalyst, respectively. This alloy catalyst also shows impressive tolerance against surface poisoning by impurity carbon monoxide (CO) in hydrogen fuel. We find that the HOR activity does not degrade obviously after 20 h of operation. The high reactivity is obtained by the optimized Ni–molybdenum (Mo) alloy nanostructure and surface that offer synergistic optimization for the adsorption of hydrogen on Ni and hydroxyl on adjacent Mo (tungsten (W)). Similarly designed WNi$_4$ alloy also demonstrates marked HOR activity in alkaline environments. Our results thus suggest a promising alloy design

strategy for producing active and durable HOR catalysts for low-cost HEMFC anodes.

## Results

**Synthesis and characterization of Ni–Mo/W.** We designed HOR catalysts on the basis of Ni and Mo (W) because they were thought to be essential elements of hydrogenase enzymes[37,38] and because Ni-based compounds have been observed to mediate the HOR catalysis in alkali with mild rates[31,34]. We first synthesized the sheet-like Mo (W)-doped Ni(OH)$_2$ precursors through microwave heating of Ni(NO$_3$)$_2$·6H$_2$O and (NH$_4$)$_6$Mo$_7$O$_{24}$·4H$_2$O (or (NH$_4$)$_{10}$W$_{12}$O$_{41}$·xH$_2$O) in a NH$_3$·H$_2$O/ethylene glycol/H$_2$O mixture at 200 °C (Fig. 1a and Supplementary Figs. 1 and 2). The resultant green powders were then annealed in hydrogen/argon (H$_2$/Ar: 5/95) atmosphere at 400 °C to produce Mo–Ni alloy (or 500 °C for W–Ni alloy; Fig. 1a). Our microwave reactor equipped with an automatic arm enables us to gain multigram-scale Mo (W)–Ni alloys in one batch (insets in Fig. 1b, e), implying a potentially large-scale use. Both two obtained alloys reveal similar morphologies when imaged by scanning electron microscopy (SEM) (Fig. 1b, e). More slit-like pores that were generated by aggregation of nanosheets during annealing process are seen for Mo–Ni alloy (Fig. 1b). Aberration-corrected high-angle annular dark-field scanning transmission electron microscopy (HAADF-STEM) of Mo–Ni alloy shows interconnected nanosheets with porous surfaces (Fig. 1c and Supplementary Fig. 3), whereas the W–Ni alloy was formed as an aggregation of overlapping nanoparticles (Fig. 1f and Supplementary Fig. 4). The morphological features yield Brunauer–Emmett–Teller (BET) surface areas of 63.3 and 33.9 m$^2$ g$^{-1}$ for Mo–Ni and W–Ni alloys (Supplementary Fig. 5), respectively. Atomic-resolution HAADF-STEM images with corresponding fast FT analyses demonstrate the formation of tetragonal MoNi$_4$ and WNi$_4$ crystalline phases (Fig. 1d, g, insets). Abundant atomic steps on the surface of MoNi$_4$ and WNi$_4$ alloys can be observed (see white arrows in Fig. 1d, g), probably induced by the high-temperature annealing treatments. X-ray diffraction (XRD) studies further confirm the successful transformation of Mo(W)-doped Ni(OH)$_2$ precursors (Supplementary Fig. 1) into fully alloyed tetragonal MoNi$_4$ (JCPDS 65-5480) and WNi$_4$ (JCPDS 65-2673) phases (Fig. 1h; corresponding crystal structures are shown as insets). It is noted that the diffraction peaks of MoNi$_4$ and WNi$_4$ differ from those of pure Ni synthesized by the same route (Fig. 1h and Supplementary Fig. 6), suggesting the alloy-induced structural change that might tune the catalytic functions. Energy-dispersive X-ray (EDX) spectrum elemental mapping in Fig. 1i presents a uniform spatial distribution of Mo(W) and Ni in Mo(W)Ni$_4$ products, and the overall Mo(W) to Ni ratio was determined to be 1:4 on the basis of EDX and inductively coupled plasma atomic emission spectroscopy (ICP-AES) measurements (Supplementary Fig. 7 and Supplementary Table 1).

The X-ray absorption spectroscopy is used to probe the impact of alloying Mo(W) on the Ni chemical environment. Figure 2a presents the X-ray absorption near-edge structure (XANES) spectra of MoNi$_4$ and WNi$_4$ at Ni K-edge, which are similar to those of the freshly synthesized Ni and Ni foil references, but greatly differ from that of NiO reference, indicating the metallic nature of the alloyed products. The radial structure function around Ni was determined by Fourier transform (FT) of extended X-ray absorption fine-structure (EXAFS) spectra (Fig. 2b and Supplementary Fig. 8). We associated the major peak at ~2.2 Å with Ni–Mo(Ni) and Ni–W(Ni) bonds in MoNi$_4$ and WNi$_4$ alloys[39,40]. No Ni–O and Ni–Ni bonds belonging to NiO reference have been observed. The decrease in peak intensity as compared to Ni–Ni bonds in freshly synthesized Ni and Ni foil

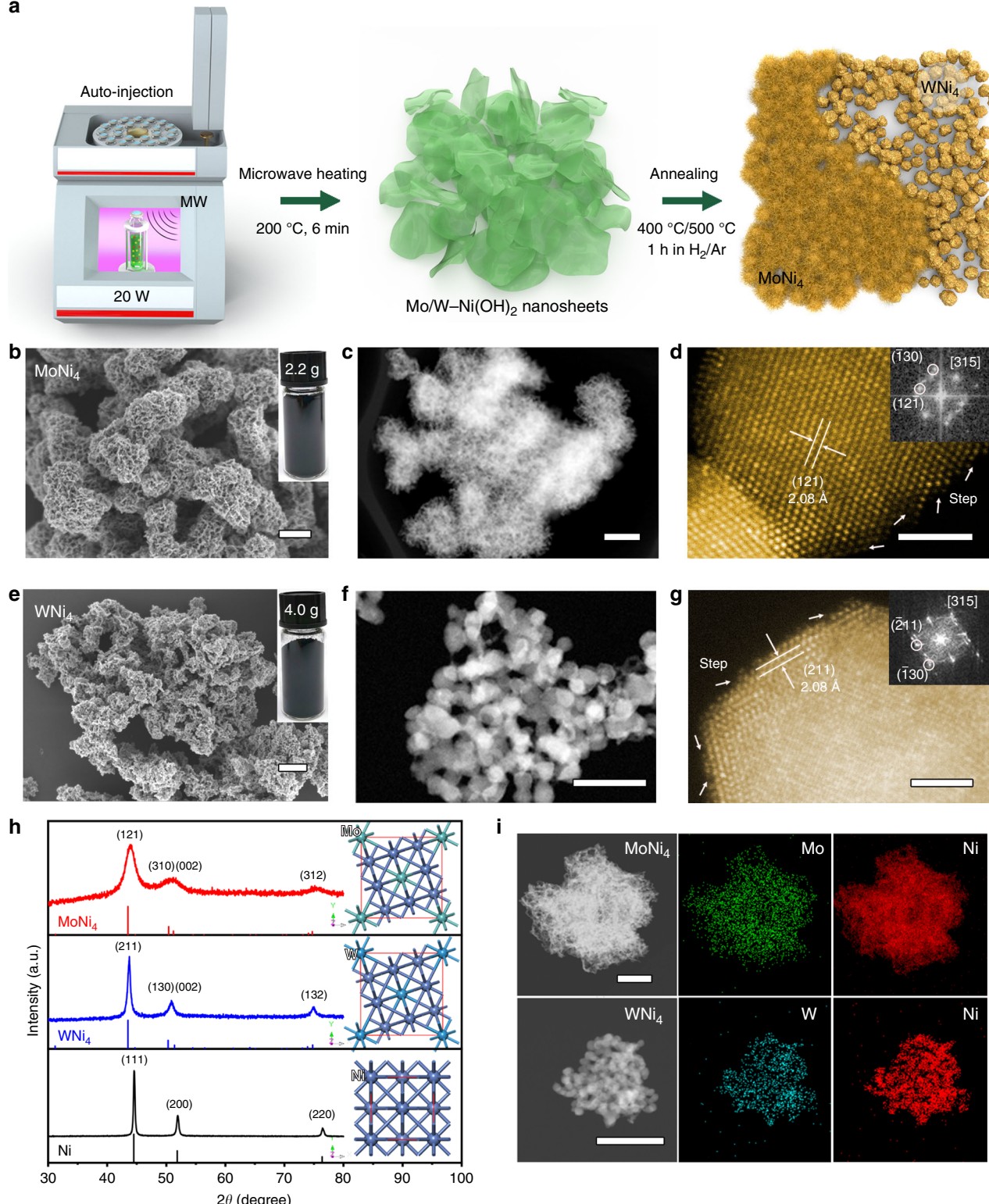

**Fig. 1 Synthesis and physical characterization of MoNi₄ and WNi₄ alloys. a** Schematic illustration of the synthesis of MoNi₄ and WNi₄ alloys. **b, e** SEM images of MoNi₄ and WNi₄ alloys, respectively. Scale bars, 200 nm. Insets in **b, e** are photographs of MoNi₄ (2.2 g) and WNi₄ (4.0 g) alloys synthesized in one batch. **c, f** STEM images of MoNi₄ and WNi₄ alloys, respectively. Scale bars, 200 nm **c** and 50 nm **f**. **d, g** Atomic-resolution HAADF-STEM images of typical MoNi₄ and WNi₄ particles, respectively. Scale bars, 2 nm. Insets in **d, g** show corresponding FFT patterns. The white arrows denote atomic steps. **h** XRD patterns of MoNi₄, WNi₄, and freshly synthesized Ni. Insets give corresponding crystal structures. **i** STEM-EDX elemental mappings of MoNi₄ and WNi₄ alloys. Scale bars, 500 nm (up) and 100 nm (down).

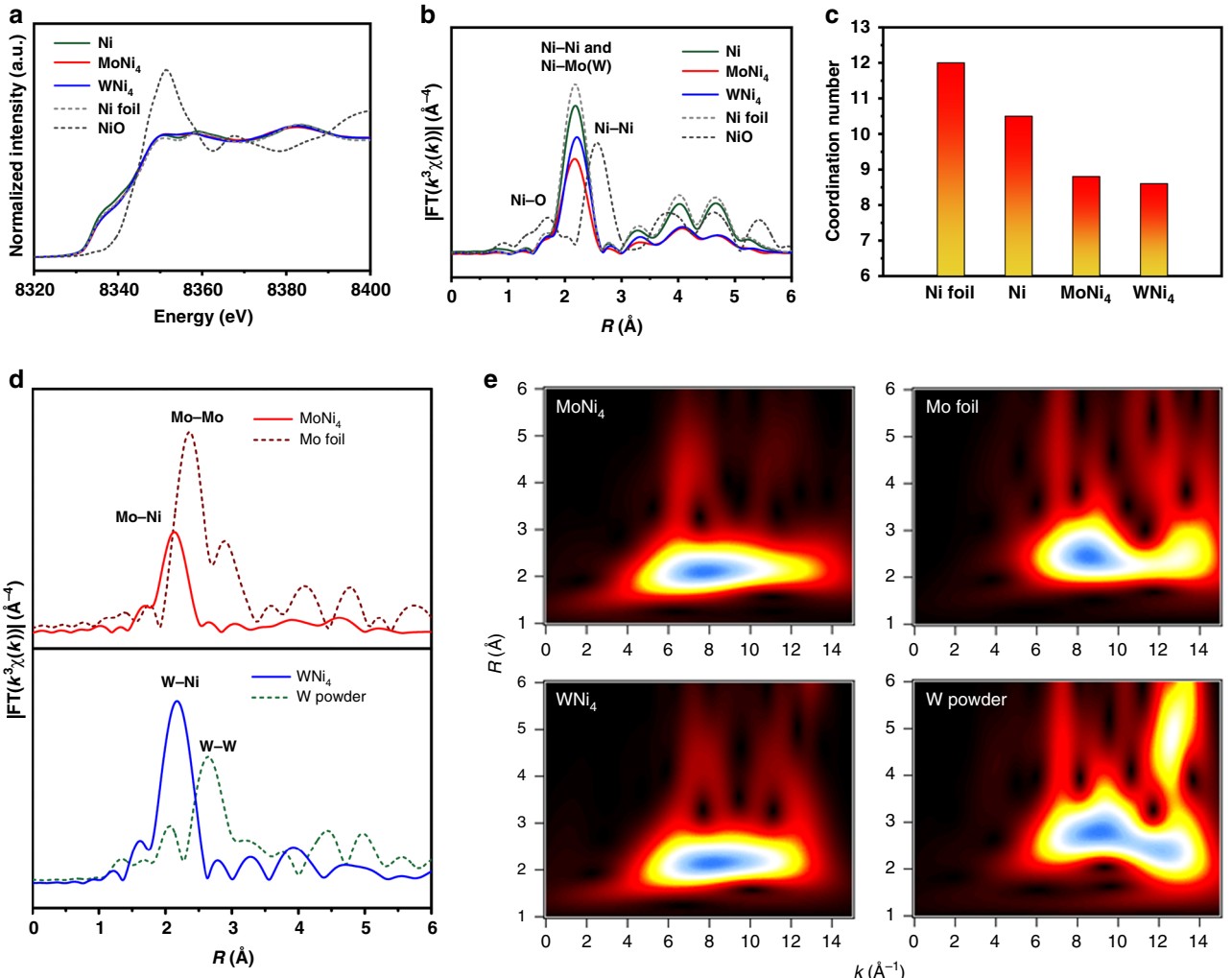

**Fig. 2 Structural analyses of MoNi$_4$ and WNi$_4$ alloys. a, b** Ni K-edge XANES spectra and corresponding Fourier transforms of $k^3$-weighted EXAFS spectra for Mo(W)Ni$_4$ alloys, freshly synthesized Ni, Ni foil, and NiO reference. **c** The average coordination numbers (CN) in the first coordination shell of Ni atoms for MoNi$_4$, WNi$_4$, and freshly synthesized Ni by EXAFS spectra curve fitting. The CN of bulk Ni (Ni foil) is 12. **d** Fourier transforms of $k^3$-weighted EXAFS spectra of Mo K-edge (up) and W L$_3$-edge (down), respectively. **e** Corresponding wavelet transforms of $k^3$-weighted EXAFS spectra of Mo K-edge (up) and W L$_3$-edge (down), respectively.

indicates the damped coordination structure of Ni. In addition, the Ni K-edge EXAFS fittings show that the first-shell Ni–Mo(W) coordination numbers (CNs) decrease from Ni (~10.5) to MoNi$_4$ (~8.8) and WNi$_4$ (~8.6), respectively. (Fig. 2c, Supplementary Fig. 9, and Supplementary Table 2). The lower CN could be attributed to the rich surface steps on our alloyed catalysts (Fig. 1d, g), which may increase the active sites that modulate the adsorption capability. The FT curves at Mo K and W L$_3$ edges in Fig. 2d show predominant peaks at ~2.2 Å, corresponding to Mo–Ni and W–Ni bonds (Supplementary Figs. 10 and 11), respectively. The results from EXAFS wavelet transform (Fig. 2e) —a powerful technique that can discriminate the backscattering atoms[41]—exhibit only one intensity maximum at ~8.0 Å$^{-1}$ in $k$ space that corresponds to Mo–Ni and W–Ni bonds in MoNi$_4$ and WNi$_4$ alloys. By contrast, wavelet transform analysis of Mo foil and W powder references give higher intensity maximum (Fig. 2e), suggesting that Mo/W atoms are forming structures in which their first coordination shell is formed only by Ni atoms and no local Mo and W metals generate in the prepared alloys, in agreement with the results in Fig. 2d. In addition, the X-ray photoelectron spectroscopy (XPS) analyses indicate a superior surface passivation resistance of our alloyed catalysts as compared

to single Ni (Supplementary Fig. 12). Together, we conclude that alloying Ni with Mo(W) cerates clear compositional and structural modulations, which we expect to benefit the HOR catalysis in alkaline electrolytes.

**Electrocatalytic HOR in alkaline electrolytes**. We now examine the electrocatalytic activity of the MoNi$_4$ and WNi$_4$ catalysts toward HOR in H$_2$-saturated 0.1 M KOH electrolyte in a three-electrode setup; with reference measurements of freshly synthesized Ni as well as commercial Pt/C (20 wt% Pt on Vulcan XC72R carbon) for comparison (see "Methods"). A very slow sweep rate of 0.5 mV s$^{-1}$ was selected to minimize the capacitance contribution and to guarantee a steady-state measurement (Supplementary Fig. 13). The optimal catalyst loading on inert glassy carbon rotating-disk electrode (RDE) was experimentally determined to be 0.5 mg cm$^{-2}$ (Supplementary Fig. 14). We note that the electrochemical data reported here were $iR$-corrected ($i$, current; $R$, resistance) for the uncompensated Ohmic drop (Supplementary Fig. 15).

Polarization curves for the HOR on studied catalysts are given in Fig. 3a, which show that MoNi$_4$ and WNi$_4$ catalysts possess an

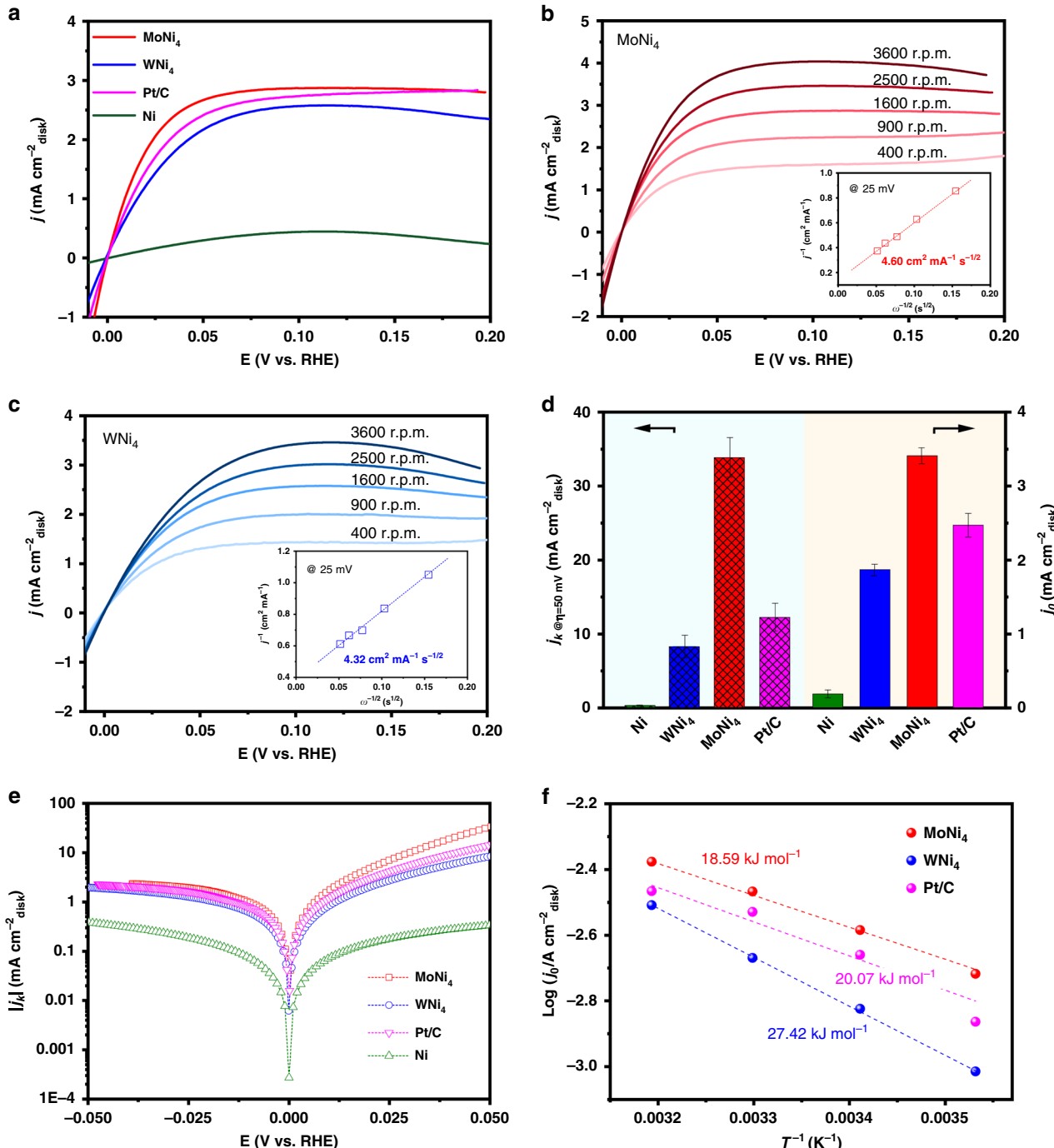

**Fig. 3 Electrocatalytic HOR performances. a** Polarization curves for the HOR on MoNi$_4$, WNi$_4$, freshly synthesized Ni, and commercial Pt/C catalyst measured in H$_2$-saturated 0.1 M KOH. Sweep rate: 0.5 mV s$^{-1}$. Rotation speed: 1600 r.p.m. **b**, **c** HOR polarization curves for MoNi$_4$ and WNi$_4$ alloys at various rotation speeds, respectively. Sweep rate: 0.5 mV s$^{-1}$. Insets in **b**, **c** show corresponding Koutecky–Levich plots at an overpotential of 25 mV. **d** Comparison of kinetic current density ($j_k$) at 50 mV (patterned) and apparent exchange current density ($j_0$; unpatterned) of different studied catalysts. The error bars (standard deviations) in **d** are calculated based on three independent test results. **e** HOR/HER Tafel plots of the kinetic current density on MoNi$_4$, WNi$_4$, Ni, and Pt/C in H$_2$-saturated 0.1 M KOH. **f** Arrhenius plots of the HOR/HER exchange current densities on MoNi$_4$, WNi$_4$, and Pt/C catalysts in 0.1 M KOH.

onset potential for yielding HOR current as low as 0 V versus the reversible hydrogen electrode (RHE), indicating their remarkable energetics for HOR in alkali. By contrast, the freshly synthesized Ni catalyst affects only negligible HOR activity. The two PGM-free HOR alloy catalysts can reach the diffusion-limiting current in the potential region >0.05 V, while a mixed kinetic-diffusion control region happens between 0 to 0.05 V. Figure 3a also reveals

that the MoNi$_4$ catalyst even outperforms the state-of-the-art Pt/C catalyst from the kinetic to the diffusion-limiting regions. The half-wave potential for MoNi$_4$ catalyst at 1600 r.p.m. is mere 14 mV, which is ~5 mV lower than that of the Pt/C catalyst, further evidencing excellent HOR activity of the MoNi$_4$ catalyst.

We then studied the HOR polarization curves on our MoNi$_4$ and WNi$_4$ catalysts as a function of the rotation rate, where the

plateau current density grows with increasing rotation rate due to the promoted mass transport (Fig. 3b, c). The Koutecky–Levich plots constructed at 25 mV show a linear relationship between the inverse of overall current density and the square root of the rotation rate, which yield calculated slopes of $4.60$ cm$^2$ mA$^{-1}$ s$^{-1/2}$ for MoNi$_4$ (inset in Fig. 3b) and $4.32$ cm$^2$ mA$^{-1}$ s$^{-1/2}$ for WNi$_4$ (inset in Fig. 3c), reasonably matching with the theoretical value of $4.87$ cm$^2$ mA$^{-1}$ s$^{-1/2}$ for the two-electron HOR process[34]. We further used Koutecky–Levich equation to calculate the kinetic current density ($j_k$). At 50 mV, a geometric $j_k$ of $33.8$ mA cm$^{-2}$ was obtained for MoNi$_4$ catalyst, which represents 105- and 2.8-fold increase compared with those of freshly synthesized Ni and commercial Pt/C catalysts (Fig. 3d).

Next, we extracted the exchange current density ($j_0$) on studied catalysts from linear fitting of micropolarization regions ($-5$ to 5 mV; Supplementary Fig. 16). The MoNi$_4$ catalyst shows a geometric $j_0$ of $3.41$ mA cm$^{-2}$, far higher than $0.19$ mA cm$^{-2}$ for the freshly synthesized Ni and $2.47$ mA cm$^{-2}$ for the Pt/C catalyst (Fig. 3d and Supplementary Table 3). The geometric $j_0$ on WNi$_4$ catalyst is $1.87$ mA cm$^{-2}$, which is slightly lower than that of Pt/C catalyst. These values are in good agreement with the fitting results of Butler–Volmer equation in the Tafel regions (Fig. 3e; see "Methods" for details). Intrinsic HOR activities of studied catalysts were further compared by the electrochemical active surface area (ECSA)-normalized $j_0$ (for details see "Methods" and Supplementary Figs. 17 and 18). The MoNi$_4$ and WNi$_4$ catalysts deliver very high ECSA-normalized $j_0$ of $0.065$ and $0.068$ mA cm$^{-2}$, respectively, which, to the best of our knowledge, has not been achieved by any other PGM-free catalysts in alkaline electrolytes, including various Ni-based compounds synthesized by other methods (Supplementary Fig. 19 and Supplementary Table 4).

In addition, we probed the activation energy ($E_a$) of the HOR on studied catalysts via plotting geometric $j_0$ with the inverse of temperature (Fig. 3f). It has been found that a linear relationship between 283 and 313 K follows the Arrhenius behavior, from which $E_a$ values of 18.59, 27.42, and 20.07 kJ mol$^{-1}$ were obtained for the MoNi$_4$, WNi$_4$, and Pt/C catalysts (Fig. 3f and Supplementary Figs. 20 and 21), respectively. We note that the $E_a$ of 20.07 kJ mol$^{-1}$ measured for Pt/C catalyst matches reasonably with 23 kJ mol$^{-1}$ for Pt(110) reported previously[25]. The considerably smaller $E_a$ values achieved for MoNi$_4$ catalyst suggest marked kinetics for HOR in alkaline environments, even outperforming the Pt/C benchmark. Moreover, we conducted a series of control experiments and disclosed that the Mo(W) to Ni ratio and the annealing temperature are critical to the HOR activity (Supplementary Figs. 22 and 25). These experiments show that Mo–Ni and W–Ni alloys with Mo(W) to Ni ratios of 1:4 obtained by annealing at 400 and 500 °C, respectively, would lead to the best HOR performances. We further note that our MoNi$_4$ sheets outperform conventional MoNi$_4$ nanoparticles for HOR owing to the porous structure that offers rich active sites (Supplementary Fig. 26).

The results above demonstrate exceptional HOR catalysis in alkaline electrolytes on MoNi$_4$ and WNi$_4$ catalysts, from which the reactivity of MoNi$_4$ even exceeds that of Pt/C benchmark. Besides activity, another very important factor for future HEMFC anode applications is electrochemical and operating stabilities. We therefore conducted aggressive long-term stability measurements on the MoNi$_4$ and WNi$_4$ catalysts (Fig. 4). First, we performed accelerated stability tests by applying linear potential scans between 0.05 and 0.15 V at 100 mV s$^{-1}$ in H$_2$-saturated 0.1 M KOH electrolytes at room temperature. At 50 mV overpotential, the HOR current density for MoNi$_4$ catalyst shows a small loss of $0.25$ mA cm$^{-2}$ after 2000 cycles (Fig. 4a), versus a loss of $0.41$ mA cm$^{-2}$ for WNi$_4$ catalyst (Fig. 4b). Second, the

studied catalysts were deposited onto carbon papers (catalyst loading: 2 mg cm$^{-2}$) and assessed the operational stability by means of chronoamperometry ($j \sim t$). Figure 4c shows that the current density at 60 mV is stable without noticeable decay during a 20 h continuous test for MoNi$_4$ catalyst, whereas no HOR current was generated in Ar-saturated 0.1 M KOH at the same overpotential (Supplementary Fig. 27). Multiple "post-mortem" characterizations display that the morphology and structure of MoNi$_4$ catalyst are well maintained (Supplementary Figs. 28–30). By contrast, Pt/C catalyst undergoes a marked degradation, which retains a mere 57% of its original current density after 20 h operation. This large drop could be ascribed to the gradual agglomeration of Pt nanoparticles during the stability test (Supplementary Fig. 31). The WNi$_4$ catalyst also shows degradation, but at a much slower manner in comparison to the Pt/C catalyst (Fig. 4c and Supplementary Figs. 28, 29, 32). These results thus suggest that our MoNi$_4$ and WNi$_4$ catalysts have stability much better than the commercial Pt/C catalyst.

**CO-tolerance evaluation**. In fuel cells, PGM catalysts (especially Pt) at the anodes are readily poisoned by impurity gas such as CO that existed in hydrogen fuel. Such poisoning is caused by the preferential CO binding on Pt, which consequently blocks the sites for hydrogen adsorption and dissociation. Unexpectedly, we observed that the MoNi$_4$ catalyst shows exciting HOR activity even in the presence of 20,000 p.p.m. CO (Fig. 4d). At the same CO concentration, no HOR activity on the Pt/C catalyst was detected, suggesting complete poisoning of the sites for H$_2$ oxidation by CO binding (Fig. 4d). Our density functional theory (DFT) calculations exhibit a significant higher CO adsorption ability of Pt as compared to the MoNi$_4$ alloy (Supplementary Fig. 33), leading to the surface of Pt covered by CO and thus deactivation. Moreover, the preferable OH adsorption on MoNi$_4$ surface assists in the oxidation of CO adsorbed, which also explains its notable CO-tolerance property. The high CO tolerance of our MoNi$_4$ alloy catalyst could further affect the quest for designing advanced fuel cell anodes based on PGM-free materials.

**HOR enhancement mechanism**. Although Ni and Ni-based materials have been extensively studied as PGM-free HOR catalysts in alkaline electrolytes, almost all previous Ni-based HOR catalysts demonstrate a relatively low level of activity and their long-term stability is also problematic. Here, the superb HOR catalytic capability observed on our readily made MoNi$_4$ and WNi$_4$ alloys prompted us to investigate the intrinsic mechanism of the high performances, thus offering a guide for the design of more advanced HOR catalysts.

We studied the electronic structures of MoNi$_4$, WNi$_4$, and freshly synthesized Ni by using ultraviolet photoelectron spectroscopy). From the valence band spectra, we found that all these catalysts have electronic bands across the Fermi level ($E_F$; Fig. 5a). The peaks located between 0 and 2 eV could be ascribed to the metal $d$-band[42,43], which reaches the maximum at 0.28, 0.80, and 1.09 eV for freshly synthesized Ni, MoNi$_4$, and WNi$_4$, respectively. The metal $d$-band maximum with respect to the $E_F$ in our MoNi$_4$ and WNi$_4$ alloys shifted far away relative to the Ni reference. According to the $d$-band theory, these results suggest that the filling of metal-H antibonding states above $E_F$ is improved for MoNi$_4$ and WNi$_4$ catalysts, affording them a weaker adsorption energy as compared to Ni[44]. Despite the HOR mechanism on PGMs in alkaline environments is still under debate, previous studies have proposed that the low HOR activity on Ni catalyst was caused by the too high HBE[34]. Thus, we partially attribute the notable HOR reactivity observed on MoNi$_4$ and WNi$_4$ catalysts to the appropriately weakened HBEs.

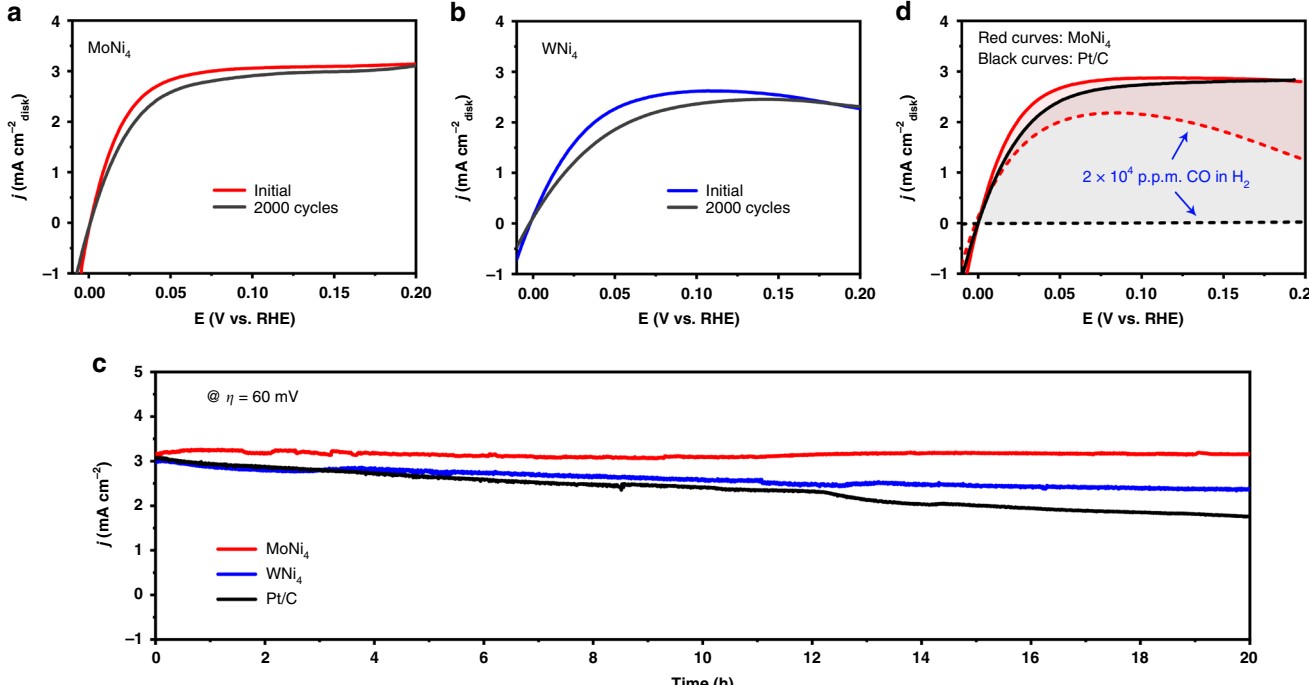

**Fig. 4 Performance stability and CO resistance. a, b** HOR polarization curves for MoNi₄ and WNi₄ alloys in H₂-saturated 0.1 M KOH before and after 2000 cycles, respectively. **c** Chronoamperometry (j – t) responses recorded on MoNi₄, WNi₄, and Pt/C catalysts at a 60 mV overpotential in H₂-saturated 0.1 M KOH. **d** HOR polarization curves for MoNi₄ and Pt/C alloys in H₂-saturated 0.1 M KOH with (dashed lines) and without (solid lines) the presence of 20,000 p.p.m. CO.

We note that OHBE, or both HBE and OHBE, are also thought to be the activity descriptor for HOR catalysis, which have recently been theoretically predicted and experimentally explored[9,27], such as the study on PdCu nanocatalyst for alkaline HOR[9]. Considering that CO can adsorb specifically on many metal surfaces[10,45,46], we thus performed CO-stripping experiments to monitor the OH binding on our catalyst surface because $OH_{ad}$ facilitates the removal of $CO_{ad}$[21]. Results of our CO-stripping experiments shown in Fig. 5b reveal that CO-stripping peak on Pt/C catalyst locates at 0.69 V, consistent with the previous reports[23]. Figure 5b further shows lower CO-stripping peaks at 0.52 and 0.49 V for MoNi₄ and WNi₄ catalysts, whereas the freshly synthesized Ni exhibits negligible CO-stripping peak. Some prior works have ascribed the sluggish HOR kinetics on Pt to its weak $OH_{ad}$ binding in alkali[8,23,27]. Our CO-stripping results here suggest that the enhanced OHBE on MoNi₄ and WNi₄ could also be responsible for their high HOR energetics.

To better understand the mechanism underlying the notable HOR performance, DFT calculations were further performed. We created and optimized catalyst models of MoNi₄(211), WNi₄(211), Ni(111), and Pt(111) to represent the catalytic surfaces (see "Methods"; Supplementary Figs. 34–36). The DFT results show that the HBE of Ni(111) is too strong, while MoNi₄(211) gives a very similar HBE with Pt(111) model (Fig. 5c–f). When comparing the OHBE of MoNi₄(211) with that of Pt(111), we observed a greatly promoted hydroxyl adsorption, which can explain the superior HOR reactivity of MoNi₄ catalyst (Fig. 5c–f and Supplementary Table 3). Our computational results suggest that alloying Ni with Mo(W) not only weakens the HBE on Ni sites but also permits an enhanced OHBE on the Mo(W) sites, which substantially promote the key Volmer step (Supplementary Figs. 37–39), leading to the HOR enhancements. We also computed the HBE and OHBE of other catalysts, such as Mo, W, CoNi₄, and FeNi₄, for comparison (Fig. 5c and Supplementary Figs. 36 and 40), and more

calculation information are shown in Supplementary Figs. 38, 39, and 41. The simulations reveal that although these materials have stronger OHBE than Pt(111), their HBEs are unfortunately too strong (Fig. 5c), giving rise to the poor HOR activity (Supplementary Figs. 42–45).

Taken together, our UPS and CO-stripping measurements, in conjunction with DFT calculations, have proposed that a synergistic interplay between HBE and OHBE likely determines the HOR rate in alkaline electrolytes; and the striking HOR reactivity observed on MoNi₄ and WNi₄ alloys could be interpreted as the optimum adsorption of hydrogen on Ni and hydroxyl on Mo(W), thus boosting the rate-limiting Volmer reaction in alkaline HOR catalysis (Fig. 5d–f and Supplementary Fig. 46).

## Discussion

In conclusion, ultrahigh HOR activity in the alkaline electrolyte has been achieved on PGM-free bimetallic MoNi₄ and WNi₄, from which the MoNi₄ exhibits exceptional apparent exchange current density that even outperforms the commercial Pt/C catalyst. We explain such high HOR rates of our catalysts by the optimized adsorption of both hydrogen and hydroxyl species owing to a synergistic effect between Ni and Mo (W). The catalysts do not show obvious deactivation over a 20-h testing period and demonstrate a good CO-tolerant property. We anticipate that further improvement in activity would be attainable by alloying other metallic elements into a single nanocatalyst. Our results reinforce the importance of alloy design concept for obtaining high-performance PGM-free HOR catalysts for future HEMFC anodes.

## Methods

**Synthesis of MoNi₄ and WNi₄ alloys.** All chemicals were used as received without further purification. The MoNi₄ and WNi₄ alloys were synthesized through a two-step method. First, sheet-like Mo(W)-doped Ni(OH)₂ precursors were

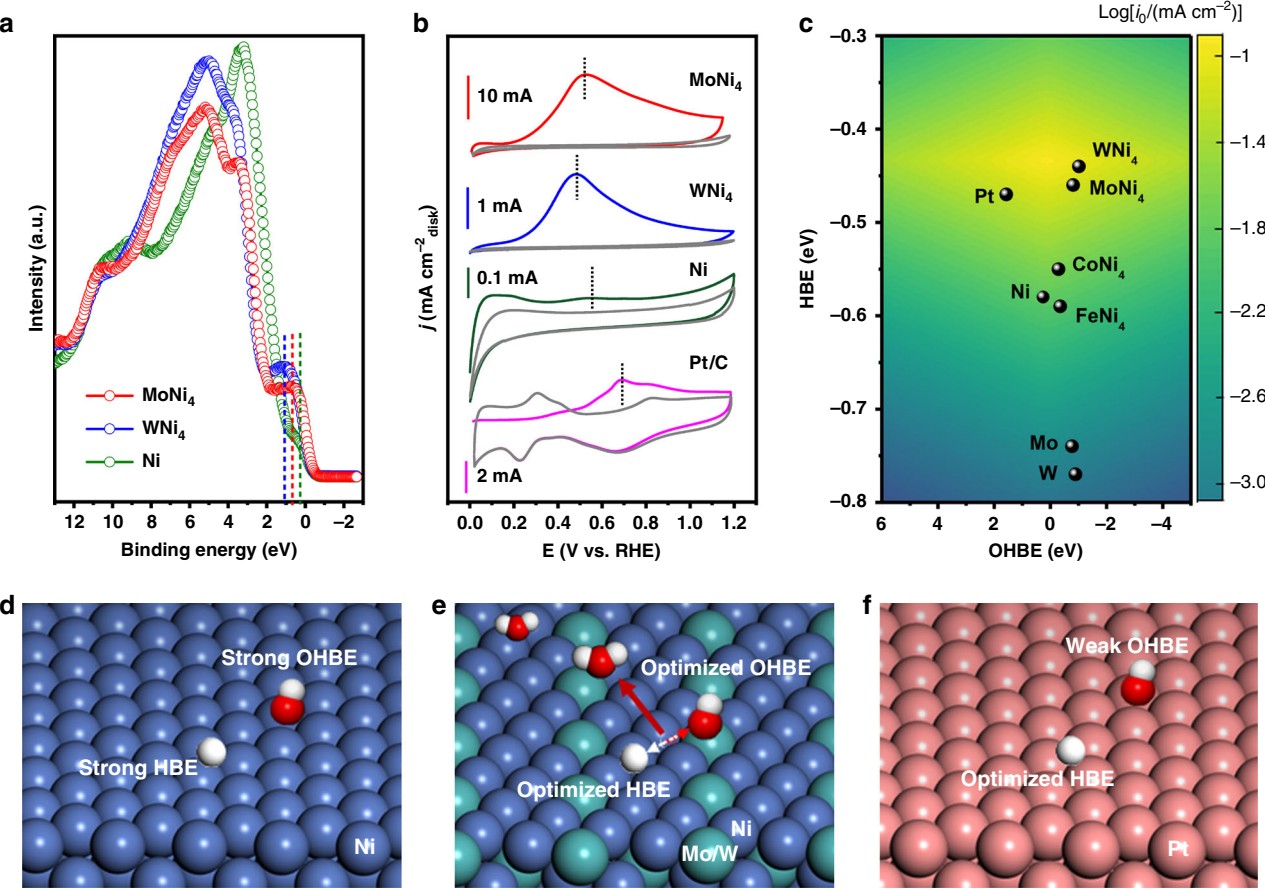

**Fig. 5 HBE and OHBE. a** UPS spectra of MoNi$_4$, WNi$_4$, and freshly synthesized Ni catalysts. **b** CO-stripping measurements on MoNi$_4$, WNi$_4$, and freshly synthesized Ni and Pt/C catalysts. Sweep rate: 20 mV s$^{-1}$. Rotation speed: 1600 r.p.m. The gray curves in **b** show the second cycle of the measurements. **c** The experimentally measured exchange current density normalized by ECSA, log($j_0$), for the HOR in 0.1 M KOH on different catalysts plotted with the calculated H and OH binding energies. **d**–**f** Schematic illustration of hydrogen and hydroxyl adsorption on freshly synthesized Ni, Mo(W)Ni$_4$ alloys, and Pt, respectively.

synthesized by microwave heating route. Briefly, 872 mg of Ni(NO$_3$)$_2$·6H$_2$O and 96 mg (NH$_4$)$_6$Mo$_7$O$_{24}$·4H$_2$O (or 190 mg (NH$_4$)$_{10}$W$_{12}$O$_{41}$·$x$H$_2$O) were dispersed into 3 mL H$_2$O in a 25 mL microwave glass vessel, followed by the addition of 1.2 mL NH$_3$·H$_2$O and 15 mL ethylene glycol. After stirring for 20 min, the mixture was irradiated in the microwave reactor (Monowave 450, Anton Paar) at 200 °C for 6 min with continuous magnetic stirring. After cooling to room temperature, the green powders were collected by centrifugation, washed, and dried for use. After that, the obtained Mo(W)-doped Ni(OH)$_2$ precursors were annealed in H$_2$/Ar (5/95) atmosphere at 400 and 500 °C for 1 h with a heating rate of 3 °C min$^{-1}$ to produce MoNi$_4$ and WNi$_4$ alloys, respectively[47].

**Synthesis of CoNi$_4$, FeNi$_4$, Ni, Mo, and W.** The CoNi$_4$ and FeNi$_4$ alloys were synthesized by the same method that was used for making MoNi$_4$ alloy, while replacing the (NH$_4$)$_6$Mo$_7$O$_{24}$·4H$_2$O with 218 mg Co(NO$_3$)$_2$·6H$_2$O and 303 mg Fe (NO$_3$)$_3$·9H$_2$O, respectively. The Ni nanopowder was synthesized through the same method for MoNi$_4$ synthesis, but without adding (NH$_4$)$_6$Mo$_7$O$_{24}$·4H$_2$O in the first step. The Mo metal was synthesized by annealing the MoO$_{3-x}$ nanorods in H$_2$/Ar (5/95) atmosphere at 800 °C for 2 h, in which the MoO$_{3-x}$ nanorods were prepared according to a previous work[48]. The W metal was synthesized by annealing the WO$_3$·H$_2$O nanosheets in H$_2$/Ar (5/95) atmosphere at 300 °C for 1 h and subsequently at 850 °C for 2 h, in which the WO$_3$·H$_2$O nanosheets were prepared according to the previous work[49].

**Characterization.** SEM (Zersss Supra 40) and TEM (Hitachi H7700) were performed to investigate the morphology of the samples. The STEM and HRTEM images, and EDX elemental mappings, were obtained on Atomic-Resolution Analytical Microscope (JEM-ARM 200F) with an acceleration voltage of 200 kV. N$_2$ adsorption/desorption analysis was taken on an ASAP 2020 (Micromeritics, USA) at 77 K. XRD was conducted on a Philips X'Pert Pro Super with Cu Kα radiation (λ = 1.541841 Å). ICP-AES results were taken by Optima 7300 DV instrument. The UPS was conducted on the BL10B beamline and XPS was

conducted on the BL11U beamline of National Synchrotron Radiation Laboratory in Hefei (China).

**XAFS measurements.** The XAFS spectra (Ni K-edge, Mo K-edge, and W L$_3$-edge) were collected at 1W1B station in Beijing Synchrotron Radiation Facility. The $k^3$-weighted EXAFS spectra were obtained by subtracting the post-edge background from the overall absorption, followed by normalizing with respect to the edge-jump step. Next, the real ($R$) space is obtained by Fourier transformation of $k^3$-weighted $\chi(k)$ data with a Hanning window (d$k$: 1.0 Å$^{-1}$) to separate the signal contributions from different coordination shells. Least-squares curve parameter fitting was executed using the ARTEMIS module of the IFEFFIT software packages[50] to study the quantitative structural parameters around the central atoms.

**Electrochemical measurements.** The HOR electrochemical measurements were conducted by a conventional three-electrode system on the electrochemical workstation (IM6ex, Zahner-Elektrik). An RDE with glassy carbon (PINE with a diameter of 5.00 mm and a disk area of 0.196 cm$^2$) was applied as the working electrode. The Ag/AgCl (3.5 M KCl) electrode and carbon rod were used as reference electrode and counter electrode, respectively. The RHE calibration was performed in high-purity H$_2$-saturated 0.1 M KOH with a Pt foil as the working electrode ($E_{RHE} = E_{Ag/AgCl} + 0.967$ V).

To make working electrodes, 10 mg catalyst powders were dispersed in 920 μL ethanol with 80 μL Nafion (5 wt%), which yield a homogeneous ink by ultrasonication. Then, 10 μl catalyst ink was pipetted onto a glassy carbon electrode, resulting in a catalyst loading of ~0.5 mg cm$^{-2}$. Before HOR measurements, 0.1 M KOH electrolyte was bubbled with high-purity H$_2$ gas for 30 min. The electrodes were pre-cycled between −1.1 and −0.75 V versus Ag/AgCl with a sweep rate of 10 mV s$^{-1}$ for 10 cycles to reach a stable state, then HOR polarization curves were collected with a sweep rate of 0.5 mV s$^{-1}$. The electrochemical impedance spectroscopy measurement was carried out at 30 mV overpotential and an amplitude of the sinusoidal voltage of 5 mV (frequency range: 100 kHz–40 mHz). For stability test, the catalysts were loaded onto clean carbon

fiber paper (catalyst loading: ~2 mg cm$^{-2}$) and used as a working electrode to perform chronoamperometry experiments at a constant potential of 60 mV versus RHE (iR free).

The kinetic current density was calculated by Koutecky–Levich equation. The measured overall HOR current density ($j$) can be divided into kinetic current density ($j_k$) and diffusion current density ($j_d$) based on the Koutecky–Levich equation:

$$\frac{1}{j} = \frac{1}{j_k} + \frac{1}{j_d},\qquad(1)$$

where $j_d$ for a rotating-disk electrode can be described by the Levich equation:

$$j_d = 0.62nFD^{3/2}\nu^{-1/6}C_0\omega^{1/2} = BC_0\omega^{1/2},\qquad(2)$$

in which $n$ is the number of electrons involved in the oxidation reaction, $F$ is the Faraday constant, $D$ is the diffusion coefficient of the reactant, $\nu$ is the viscosity of electrolyte, $C_0$ is the solubility of $H_2$ in the electrolyte, $B$ is the Levich constant, and $\omega$ is the rotating speed.

The exchange current ($j_0$) can also be obtained by fitting the linear portion of the Tafel plots, where the Bulter–Volmer equation can be converted to Tafel equation:

$$\eta = \mathrm{Log}(j_0) + b \times \mathrm{Log}(j).\qquad(3)$$

CO stripping was performed by holding the electrode potential at 0.1 V versus RHE for 10 min in the purged CO to adsorb CO on the metal surface, followed by Ar purging for another 30 min to remove residual CO in the electrolyte. The CO-stripping current was obtained via cyclic voltammetry in a potential region from 0 to 1.2 V at a sweep rate of 20 mV s$^{-1}$.

**DFT calculations**. All the computations were performed by using the Vienna ab initio Simulation Package[51] at the spin-polarized DFT level. The electronic exchange and correlation effects were described with the Perdew–Burke–Ernzerhof formalism[52] within a generalized gradient approximation. The interaction between the ion cores and valence electrons was simulated by the all-electron projector-augmented wave[53]. The exchange correction treated for the transitional metals was based on the previous literatures, which includes the same metals as ours for reliability and comparability[33,34,54]. A kinetic cut-off energy of 500 eV was employed for the plane-wave expansion and a Gaussian electron smearing of 0.1 eV was used. The (3 × 3)-Ni(111), (3 × 3)-Mo(110), (3 × 3)-W(110), (3 × 3)-Pt(111), (1 × 1)-CoNi$_4$(111), (1 × 1)-FeNi$_4$(111), (1 × 1)-MoNi$_4$(211), and (1 × 1)-WNi$_4$(211) slabs with four layers and 15 Å vacuum layer were modeled to simulate the explored surfaces of metals and alloys. The convergence criteria for the forces and energy were 10$^{-4}$ eV and 0.02 eV Å$^{-1}$, respectively, with the bottom layer fixed, while other layers relaxed during geometry optimization. These convergence criteria were chosen according to the previous literatures to ensure the accuracy[34,54]. The 9 × 9 × 9 and 5 × 5 × 1 Monkhorst–Pack k-point grids were sampled for the bulk and slab structures, separately.

The adsorption energies of hydrogen and hydroxyl species with the explored catalysts were calculated according to $E_{H\text{-ads}} = E_{H@cat.} - E_{cat.} - E_H$ and $E_{OH\text{-ads}} = E_{OH@cat.} - E_{cat.} - E_{OH}$, where $E_{H@cat.}$ and $E_{OH@cat.}$ represent the energy of metals or alloys slabs with the adsorbed hydrogen and hydroxyl species, while $E_{cat.}$, $E_H$, and $E_{OH}$ stand for the energies of the metals or alloys slabs, the hydrogen atoms, and hydroxyl species, respectively. A more negative $E_{H\text{-ads}}$ or $E_{OH\text{-ads}}$ indicates a larger binding energy.

## Data availability
The data that support the findings of this study are available from the corresponding authors upon request.

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

## Acknowledgements
We acknowledge the funding support from the National Natural Science Foundation of China (Grants 21521001, 21431006, 21225315, 21321002, 91645202, 51702312, and 21975237), the Users with Excellence and Scientific Research Grant of Hefei Science Center of CAS (2015HSCUE007), the Key Research Program of Frontier Sciences, CAS (Grant QYZDJ-SSW-SLH036), the National Basic Research Program of China (Grants 2014CB931800 and 2018YFA0702001), the Chinese Academy of Sciences (Grants KGZD-EW-T05 and XDA090301001), the Strategic Priority Research Program of the Chinese Academy of Sciences (XDA21000000), the Fundamental Research Funds for the Central Universities (WK2060190045 and WK2340000076), and the Recruitment Program of Global Youth Experts. Z.-Y.Y. acknowledges the Anhui Provincial Natural Science Foundation (Grant 1908085QB60). We acknowledge the BL10B beamline of the National Synchrotron Radiation Laboratory in Hefei (China). This work was partially carried out at the USTC Center for Micro and Nanoscale Research and Fabrication.

## Author contributions
M.-R.G., Y.D., and S.-H.Y. conceived the idea. M.-R.G. and S.-H.Y. supervised the project. Y.D., Z.-Y.Y., and C.-T.Z. performed the experiments, and collected and analyzed the data. L.Y. and J.J. carried out the DFT calculations. X.-S.Z., H.-.H.D., and J.-F.Z. performed UPS and XPS measurements. L.S. performed the HAADF-STEM measurements. L.-R.Z. and C.G. collected and analyzed the XANES data. X.-T.Y., F.-Y.G., X.-L.Z., X.Y., and R.L. assisted with the experiments and characterizations. M.-R.G., Y.D., and S.-H.Y. co-wrote the manuscript. All authors discussed the results and commented on the manuscript.

## Competing interests
The authors declare no competing interests.
