## [Peer Review File · Nature Communications]

Reviewers' comments:

Reviewer #1 (Remarks to the Author):

Major modification is suggested for the draft by Yu et al. reporting MNi₄ (M = Mo or W) catalysts for the HOR in alkaline. The major achievement of this work is the production of a group of PGM-free catalysts for the HOR with the activity comparable to that of Pt. The major weakness is the rationalization of the high HOR activity.

1. The major issue of Ni as a HOR catalyst in alkaline is that Ni is full passivated within the HOR potential region and thus has no affinity toward H₂, which is reflected by the negligible limiting HOR current density of Ni as shown in Figure 3A. Therefore, the major merit of the MNi₄ lies in the high limiting current density of ~3 mA/cm² at 2,500 rpm. The authors shall explain why the Ni remained unpassivated, or the current is delivered by Mo/W (I don't think so)? and shall at least hypothesize what are the active sites. The XAS results actually provide some hints. As seen in Figure 2a and 2b, the Ni in MoNi₄ and WNi₄ remains largely at reduced phase ex situ exposed to air. On the other hand, the surface of Ni nanoparticles is passivated ex situ.
2. The HBE and OHBE supported by UPS and DFT were routinely done without giving true reasons or insights. If both the HBE and OHBE matter, the authors shall clearly explain which site(s) dissociate H₂, which site binds OH_{ad}, and subsequently which sites promote the Volmer step (removal of H_{ad}) in which way (HOR mechanism).
3. The claim of the structure of tetragonal MoNi₄ or WNi₄ was not fully justified. The XRD and XAS results clearly indicate that the bulk structure of these two samples are reminiscent of that of Ni, but with enlarged lattice constants as expected from the Vegard's Law. The local structure can be readily determined by fitting the EXAFS data at the Mo edge. For example, if Mo is surrounded by four Ni as seen by XAS, then the authors can claim MoNi₄. The fitting at the Ni edge is confusing. Why the authors indicate Ni-Ni(Mo) or Ni-Ni(W)? Mo and W are pretty far away from Ni in the periodic table and also much bigger and thus can be easily distinguished by XAS. In addition, if the Ni-Mo or Ni-W ratio is around 4:1, the Ni-Ni bond distance shall be larger than that of Ni foil that is reflected by the shift in the XRD peaks and expected from Vegard's Law, which is however not the case here. Since the Ni-Ni bond distance is comparable to that of Ni foil, and the Mo does not see Mo but only Ni around, I suspect the samples subject to XAS measurements have a much less Mo content than MoNi₄.
4. The lack of surface characterization makes it hard to understand the exceptional HOR activity. I understand it is extremely difficult to monitor the surface composition and valence state of surface metals in situ during HOR, but ex situ surface characterization such as XPS is recommended to determine the oxidation state of Ni, Mo, and W. A comparison of the oxidation state of the surface Ni in Ni₄Mo and in Ni counterpart would be useful.
5. I don't think it is possible to accurately evaluate the ECSA of MoNi₄ for the HOR.

Reviewer #2 (Remarks to the Author):

This is an interesting work that reported nickel-molybdenum (MoNi₄) and Nickel-tungsten (WNi₄) bimetallic catalysts for hydrogen oxidation reaction (HOR) catalysis in alkaline electrolytes. The authors reported that the two nanoalloy catalysts showed very high intrinsic activities towards HOR activity compared to most of previously reported non-precious metal catalysts. The experimental results are important and sufficient characterizations, tests and computations were provided. It can be published in Nature Communications after considering the following questions.

- 1) The authors should decouple the effects of composition, structure and morphology. Since a microwave synthesis method was used to prepare high surface area of MoNi₄ and WNi₄ samples, What would the HOR activity be if they are just non-porous nanoparticles (not the nanosheet structure)? A blank experiment on HOR activity test on MoNi₄/WNi₄ nanoparticles should be given. In addition, what is the composition effect? It seems that the authors have optimized the Mo-Ni composition, however, what is the reason behind the optimal composition? Is it because of change of

the ratio of surface Ni, of the phase structure, or of chemical state of Ni, or some other reasons?

2) The authors provided the HOR durability tests, and showed both catalysts are better than Pt/C, what is the main mechanism behind it? the authors should show some characterizations after HOR tests.

3) On the DFT computation, it is convincing that both HBE and OHBE play roles on HOR in alkaline electrolyte. Can the authors use DFT to give some insights into the CO-tolerant capability of the Mo/W-Ni nanoalloy catalysts?

4) The authors showed the geometric activity of HOR on MoNi₄ is greater than a commercial Pt/C catalyst, however, they did not show the true intrinsic activity comparison between MoNi₄ and Pt/C. Supplementary Figure.16 should include the HOR activity on state-of-the-art Pt/C catalysts. Make sure that all the HOR activities on various non-previous metal catalysts were tested under same conditions.

5) It will be more exciting to see the new bimetallic catalysts tested in a realistic fuel cell, and show they can serve as a promising non-precious anode for realistic AEMFCs.

Reviewer #3 (Remarks to the Author):

This paper demonstrated that the nickel-molybdenum nanoalloy with tetragonal MoNi₄ phase can catalyze the HOR efficiently in alkaline electrolytes. This catalyst is tolerant to carbon monoxide poisoning. DFT calculations were performed to evaluate the HBE and OHBE. The authors rationalize the good efficiency in the combination of nickel and molybdenum for optimized adsorption of intermediates. Another analysis tool used is the d-band center shift trend. Some points for minor improvement:

1. Please clarify why no XC corrections were included for transition metals.
2. Please comment on the choice of 0.02 eV/Å for forces. This may be low.
3. It is not clear to me which reaction mechanism was considered for modeling HOR. The analysis includes only the OH and H adsorption steps.

We thank all the reviewers for their valuable comments and questions that help us significantly improve the revised manuscript.

REVIEWER REPORTS:

Reviewer #1 (Remarks to the Author):

Major modification is suggested for the draft by Yu et al. reporting MNi_4 ($M = Mo$ or W) catalysts for the HOR in alkaline. The major achievement of this work is the production of a group of PGM-free catalysts for the HOR with the activity comparable to that of Pt. The major weakness is the rationalization of the high HOR activity.

Response: We are very grateful for the reviewer's high praise on the development of efficient PGM-free HOR catalysts reported in this work.

1. The major issue of Ni as a HOR catalyst in alkaline is that Ni is full passivated within the HOR potential region and thus has no affinity toward H_2 , which is reflected by the negligible limiting HOR current density of Ni as shown in Figure 3A. Therefore, the major merit of the MNi_4 lies in the high limiting current density of ~ 3 mA/cm² at 2,500 rpm. The authors shall explain why the Ni remained unpassivated, or the current is delivered by Mo(W) (I don't think so)? and shall at least hypothesize what are the active sites. The XAS results actually provide some hints. As seen in Figure 2a and 2b, the Ni in $MoNi_4$ and WNi_4 remains largely at reduced phase ex situ exposed to air. On the other hand, the surface of Ni nanoparticles is passivated ex situ.

Response: We appreciate the reviewer for his/her thoughtful comments and questions. The important contribution of this work comes from our findings that $MoNi_4$ and WNi_4 nanoalloys can catalyze the alkaline HOR comparable to Pt/C catalyst, whereas almost all previous PGM-free catalysts are readily deactivated owing to surface oxidation. We attribute the outstanding activity and stability (limiting current density of ~ 3 mA cm⁻² at 2500 rpm up to 0.2 V versus RHE) to the following reasons:

First, the alloy effect. Our single Ni sample undergoes expected surface oxidation during alkaline HOR, thus showing little HOR reactivity. When alloying with Mo and W to form $MoNi_4$ and WNi_4 alloys, remarkable HOR performances achieved. Prior research has demonstrated that alloyed structures commonly possess superior resistance to passivation as compared to single metals. For examples, Ni-Mo and Ni-W alloys have been observed to show much better passivation resistance than that of microcrystalline Ni in alkaline conditions (*J. Appl. Electrochem.* **2004**, 34, 1085-1091; *Corros. Sci.* **2011**, 53, 1066-1071), because alloys usually have stronger N-M bonds than both the N-N and M-M bonds (*Chem. Rev.* **2016**, 116, 10414-10472), which thus enable superior stability.

Second, the optimized H_{ad} adsorption efficiently consumes OH_{ad} on the catalyst surface. Our computational results uncover that the HBEs are properly weakened on our $MoNi_4$ and WNi_4 alloys as compared to single Ni. As a consequence, hydrogen can dissociate and desorb rapidly from the Ni sites of $MoNi_4$ or WNi_4 alloys, while such adsorption on single Ni metal is too strong. It is well known that alkaline HOR on the catalyst surface involves the Volmer step that H_{ad} and OH_{ad} react to generate H_2O (*Sci. Adv.* **2016**, 2, e1501602). Thus, we can image that the optimized H_{ad} adsorption on the surfaces of $MoNi_4$ or WNi_4 alloys would react and consume the absorbed OH_{ad} efficiently, which significantly prevents the surface passivation of alloys under alkaline condition.

Figure R1. Ni 2p XPS characterization for $MoNi_4$, WNi_4 and single Ni, respectively.

Third, we performed additional XPS studies of our $MoNi_4$ or WNi_4 alloys and compared them to that of single Ni. As shown in **Figure R1**, the Ni 2p XPS shows two peaks at 852.6 eV and 855.7 eV, which can be assigned to Ni^0 and Ni^{2+} , respectively (*Nat. Commun.* **2017**, 8, 15437). Our XPS analyses clearly reveal that single Ni undergoes more severe surface oxidation as compared to the $MoNi_4$ or WNi_4 alloys, consistent with XAS observations.

Furthermore, in our revised manuscript, the UPS and CO-stripping experiments, in conjunction with DFT calculations, reveal the alloy effect that allows for optimum adsorption of hydrogen on Ni sites and hydroxyl on Mo(W) sites, which substantially boost the rate-limiting Volmer step, leading to the observed HOR performance of the $MoNi_4$ and WNi_4 catalysts.

Overall, we surmise that the alloy effect not only offers robust metallic bonds, but also weakens the HBE that accelerates the removal of surface OH_{ad} species through the Volmer step. These together leads to superior surface resistance to passivation of our designed alloyed catalysts and their outstanding reactivity in alkaline environment.

2. The HBE and OHBE supported by UPS and DFT were routinely done without giving true reasons or insights. If both the HBE and OHBE matter, the authors shall clearly explain which site(s) dissociate H_2 , which site binds OH_{ad} , and subsequently which sites promote the Volmer step (removal of H_{ad}) in which way (HOR mechanism).

Response: We thank the reviewer for the comments and good suggestion. As to single Ni catalyst, It is known that hydrogen bonding energy (HBE) is too high, causing its poor HOR reactivity in alkali. Our UPS measurements reveal that the alloyed $MoNi_4$ and WNi_4 catalysts permit much weaker HBEs as compared to Ni, which is important to HOR catalysis. Moreover, previous studies have shown that Ni-based materials could be developed as HOR catalysts in alkali (*Nat. Commun.* **2016**, 7, 10141; *Angew. Chem. Int. Ed.* **2019**, 58, 14179), whereas metallic Mo and W are HOR inactive (see **Supplementary Fig. 42-43**). This suggests that Ni sites could bear good H_2 dissociation ability but Mo and W do not. On the other hand, our DFT results show optimized H_{ad} species on Ni sites (**Figure 5c** and **Supplementary Fig. 38**). On the basis of these results, we conclude that Ni sites on the $MoNi_4$ and WNi_4 catalysts are appropriate active sites for the adsorption of H_{ad} species.

Recent experimental and computational studies further uncover that hydroxyl bonding energy (OHBE) should also be the activity descriptor for HOR catalysis. Importantly, the sluggish HOR on Pt catalyst in alkaline electrolytes was attributed to its weak OH_{ad} binding. In this manuscript, our CO-stripping experiments clearly exhibit the enhanced OHBE on $MoNi_4$ and WNi_4 alloys as compared to Pt/C catalyst (**Fig. 5b** in the MS). Besides, the DFT results show that Mo and W sites have stronger OH absorption than the Ni sites (see **Figure 5c** and **Supplementary Fig. 39**). These results thus indicate that OH_{ad} species are prefer to bind on Mo and W sites.

In light of above analyses, we reasonably come to a bi-functional mechanism (*Angew. Chem. Int. Ed.* **2017**, 56, 15594) for our $MoNi_4$ and WNi_4 catalysts: Ni sites on the alloyed catalysts offer the optimum hydrogen adsorption, while Mo and W sites on the alloys enable suitable adsorption of OH species (**Figure R2a**). As a result, the optimized HBE enabled by Ni and the optimized OHBE enabled by Mo or W substantially boost the rate-limiting Volmer step (Heyrovsky–Volmer pathway), leading to the marked HOR performance of our $MoNi_4$ and WNi_4 catalysts.

To further verify this mechanism, we calculated the free energy for Volmer step of the studied catalysts. The much lower energy barriers (**Figure R2b**) of $MoNi_4$ and WNi_4 catalysts further evidence the promoted Volmer step thus the improved HOR energetics.

Figure R2. a, Schematic illustration of H and OH adsorption on Mo(W)Ni₄ alloys. b, Calculated free energy diagram for the Volmer step. The gray, green, red and white spheres represent Ni, Mo(W), O and H atoms, respectively.

We have added these new data in the **revised SI (Supplementary Figure 37)** and provided the discussion properly in the **revised MS**.

3. The claim of the structure of tetragonal MoNi₄ or WNi₄ was not fully justified. The XRD and XAS results clearly indicate that the bulk structure of these two samples are reminiscent of that of Ni, but with enlarged lattice constants as expected from the Vegard's Law. The local structure can be readily determined by fitting the EXAFS data at the Mo edge. For example, if Mo is surrounded by four Ni as seen by XAS, then the authors can claim MoNi₄. The fitting at the Ni edge is confusing. Why the authors indicate Ni-Ni(Mo) or Ni-Ni(W)? Mo and W are pretty far away from Ni in the periodic table and also much bigger and thus can be easily distinguished by XAS. In addition, If the Ni-Mo or Ni-W ratio is around 4:1, the Ni-Ni bond distance shall be larger than that of Ni foil that is reflected by the shift in the XRD peaks and expected from Vegards' Law, which is however not the case here. Since the Ni-Ni bond distance is comparable to that of Ni foil, and the Mo does not see Mo but only Ni around, I suspect the samples subject to XAS measurements have a much less Mo content than MoNi₄.

Response: We thank the reviewer for the thoughtful comments and questions. We want to address your doubts and questions from the following parts:

First, by combing the FT curves and EXAFS wavelet transform at Mo K edges, we have demonstrated the formation of Mo-Ni alloy, where the predominant peaks located at 2.2 Å is ascribed to scattering from neighboring Ni atoms present at the shorter distance than Mo-Mo distance (see **Figure 2d**). Following your comments, we further fitted the EXAFS data at the Mo edge, as shown in **Figure R4** below. Our fitting results show that the obtained first-shell Mo-Ni coordination number is about 6.9, which is smaller than the coordination number of 12 for the standard MoNi₄ crystal (**Note: the coordination number for MoNi₄ crystal is 12 but not 4**). The

obtained smaller coordination number could be resulted from the nanoscale size and rich surface steps of our porous MoNi₄ nanoalloy. This observation is analogous to a recent report (*Science* **2020**, 367, 777-781), where the NiMo alloys similarly have weak intensity for the peak at 2.2 Å.

Figure R4. The raw and fitting Fourier transform of k^3 -weighted EXAFS spectra of Mo K-edge for MoNi₄.

Second, we have to note that Mo(W) in Ni alloy are difficult to distinguish by the Ni K edge XAS. Such dilemma has also been demonstrated previously (*Science* **2020**, 367, 777-781 and *Sci. Adv.* **2017**, 3, e1603068), which means that it is hard to differentiate the Ni-Ni and Ni-Mo(W) bonds from Ni K edge XAS. Owing to this reason, we used Ni-Ni(Mo) or Ni-Ni(W) in our original manuscript. Directly estimating Ni-Ni(Mo) bond distance from the XAS spectra is inaccurate. For example, a recent work (*Science* **2020**, 367, 777-781) on MoNi alloy (atomic ratio=1: 3.49; close to MoNi₄) displays a similar position for Ni-Ni(Mo) peaks between NiMo alloy and Ni foil from the Ni K edge results (see **Figure R5**). Nevertheless, an accurate estimate of bond distance can be realized by the EXAFS fitting. Our EXAFS fitting results show a bigger Ni-Ni(Mo/W) bond distance of alloys than that of Ni metal and Ni foil (see **Supplementary Table 2**).

Figure R5. EXAFS of Mo K-edge and Ni K-edge of NiMo Catalysts reported in *Science* **2020**, 367, 777-781.

Last, we highlight that the XRD pattern of our MoNi₄ alloy can be perfectly assigned to the tetragonal MoNi₄ (*JCPDS* 65-5480) phase (slightly shifting to lower degree as compared to pure Ni). More importantly, our ICP-AES results confirm a Mo:Ni atomic ratio of ~1:4 for 10 independent measurements (**Figure R6**). Additionally, the Mo-Ni alloy phase diagrams (**Figure R7**) reveal that new MoNi₄ phase forms when the Ni:(Mo+Ni) ratio is 0.8 (the ratio is the same with our MoNi₄ alloy) and the temperature is below 1100 K (see the Red Star in **Figure R7**).

Figure R6. ICP results for different batches of MoNi₄ alloys.

Figure R7. Alloy phase diagrams of Mo-Ni from FactSage® thermochemical values collection on SGTE 2017 alloy database.

Together, our above results and analyses could reasonably verify tetragonal MoNi₄ phase we have obtained. We further note that MoNi₄ alloy phase has also been synthesized and reported previously, such as *Nat. Commun.* **2017**, 8, 15437 and *Adv. Mater.* **2017**, 29, 1703311.

4. The lack of surface characterization makes it hard to understand the exceptional HOR activity. I understand it is extremely difficult to monitor the surface composition and valence state of surface metals in situ during HOR, but ex situ surface characterization such as XPS is recommended to determine the oxidation state of Ni, Mo, and W. A comparison of the oxidation state of the surface Ni in Ni₄Mo and in Ni counterpart would be useful.

Response: We thank the reviewer for the good comments and suggestion. The inevitable exposure of the catalysts in air causes surface oxidation, which largely hampers the examination on the original surfaces of catalysts. As the reviewer mentioned, *in-situ* studies will eliminate the surface oxidation issue, but current techniques can not allow us (also for other research groups) to monitor the dynamic surface process during the rotating disk electrode testing. Hence, we try the best to avoid the air exposure of our catalysts and studied their surface chemistry before and after HOR process by XPS characterizations.

As to the Ni 2p XPS spectra (**Figure R8a**), the poignant peaks at 852.6 eV and 869.8 eV are assigned to Ni⁰, while the weak peaks located at 855.7 eV and 873.5 eV are indexed to Ni²⁺ (*Nat. Commun.* **2017**, 8, 15437). For Mo 3d XPS spectra (**Figure R8b**), the peaks located at 227.9 eV, 231.1 eV, 232.1 eV and 235.3 eV can be indexed to Mo⁰ 3d_{5/2}, Mo⁰ 3d_{3/2}, Mo⁴⁺ 3d_{3/2} and Mo⁶⁺ 3d_{3/2}, respectively (*Nat. Commun.* **2017**, 8, 15437). For W 4f XPS spectra in **Figure R8c**, the peaks located at 31.3 eV, 33.5 eV, 35.3 eV, and 37.5 eV are indexed to W⁰ 4f_{7/2}, W⁰ 4f_{5/2}, W⁶⁺ 4f_{7/2} and W⁶⁺ 4f_{5/2}, respectively (*ACS Appl. Nano Mater.* **2018**, 1, 1228-1235). Although with great care during XPS measurements, these samples are still inevitably exposed in air for a certain time, causing some surface oxidation of catalysts, agreeing with previous XPS studies on alloys (*Small* **2017**, 13, 1701648). Even so, our XPS results also reveal that single Ni metal possesses much higher Ni²⁺ peak than MoNi₄ and WNi₄ alloys (**Figure R8a**), demonstrating that the surfaces of MoNi₄ and WNi₄ alloys are more difficult to be oxidized owing to the alloy effect, which would facilitate HOR catalysis.

Figure R8. XPS Characterization. **a**, Ni 2p XPS of MoNi₄, WNi₄ and freshly-synthesized Ni, respectively. **b**, Mo 3d XPS of MoNi₄. **c**, W 4f XPS of WNi₄.

After the HOR, the cycled catalysts were carefully collected and used for XPS examinations again. As demonstrated in **Figure R9a**, we find that the MoNi₄ and WNi₄ alloys still have strong Ni⁰ signals even after the long-term stability test. The Ni²⁺ signals come from their surface oxidation when exposing in air and in alkali. By contrast, Ni⁰ signals almost disappeared for single Ni catalyst, with only Ni²⁺ signals left. These results unambiguously reveal that our designed alloyed catalysts bear more robust surface structures that protect HOR active sites than single Ni metal catalyst when performing HOR in alkaline environments. Moreover, the Mo 3d and W 4f XPS spectra (**Figure R9b and c**) both only show slight change, suggestion the marked robustness of the MoNi₄ and WNi₄ alloyed catalysts.

Figure R9. XPS characterization after HOR test for 3h. **a**, Ni 2p XPS of MoNi₄, WNi₄ and Ni after HOR test. **b**, and Mo 3d XPS of MoNi₄ after HOR test. **c**, W 4f XPS of WNi₄ after HOR test.

We have added these new data in the **revised SI (Supplementary Figure 12, 29)** and provided some discussion properly over there and the **revised MS**.

5. I don't think it is possible to accurately evaluate the ECSA of MoNi₄ for the HOR.

Response: We thank the reviewer for this insightful comments.

We know that the best way to evaluate the intrinsic electrochemical activity of a catalyst is to calculate its specific activity based on its electrochemically active surface area (ECSA). However, the ECSA values are difficult to obtain for many non-noble metals: they cannot be calculated using the classic hydrogen under-potential deposition (UPD) like commonly done for Pt because no obvious hydrogen adsorption occurs prior to H₂ evolution. Alternatively, for non-noble catalysts, we have to turn to other methods to evaluate the ECSAs.

Commonly, the ECSA can also be quantified by the redox reaction of surface metals, which relies on the interaction between the surface metal atoms and oxygenated species. In the original manuscript, we evaluated the ECSAs of Ni-based catalysts from the OH desorption region using a charge density of 514 $\mu\text{C cm}^{-2}_{\text{Ni}}$ for one monolayer of OH adsorption. This method has recently been widely adopted to evaluate ECSA values of Ni-based catalysts, such as Ni/N-CNT (*Nat. Commun.* **2016**, 7, 10141), Ni/CeO₂ (*Angew. Chem. Int. Ed.* **2019**, 58, 14179-14183), Ni₃N (*Angew. Chem. Int. Ed.* **2019**, 58, 7445-7449), CoNiMo (*Energy Environ. Sci.* **2014**, 7, 1719-1724), etc. Therefore, we believe that the method we used to evaluate the ECSA values of our studied catalysts is reasonable, which could offer a fair comparison of our developed catalysts with that of other HOR catalysts reported previously.

Figure R10. ECSA by non-faradaic double layer capacitance obtained in CH₃CN with 0.15 M KPF₆. **a**, CV curves of MoNi₄ alloy collected at various scan rates ranging from 10 to 50 mV s⁻¹. **b**, The corresponding linear fitting of scan rates versus difference between the anodic and cathodic current at -0.15 V vs. Ag/AgCl.

Motivated by your comments, we double checked the ECSA values of our catalysts from the double-layer capacitance according to equation $\text{ECSA} = C_{\text{dl}}/C_s$ (where C_s is the specific capacitance of the catalyst or the capacitance of smooth planar surface per unit area) that suggested by McCrory, Peters and Jaramillo (*J. Am. Chem. Soc.* 2013, 135, 16977). For the estimate of surface area, we

adopted the general specific capacitance of $C_s = 11 \mu\text{F cm}^{-2}$ based on typical reported value (*Angew. Chem. Int. Ed.* **2019**, 58, 10644-10649). As shown in **Figure R10**, our measurements give an ECSA value of 10.64 cm^2 for the MoNi_4 alloy loaded on the glassy carbon electrode (0.196 cm^2), matching well with our previous result of 10.32 cm^2 . Thus, for our PGM-free catalysts, we could gain relatively accurate ECSA values for the comparison of their intrinsic HOR activities.

We have added the new data in the **revised SI (Supplementary Figure 18)** and provided some details over there.

Reviewer #2 (Remarks to the Author):

This is an interesting work that reported nickel-molybdenum (MoNi_4) and Nickel-tungsten (WNi_4) bimetallic catalysts for hydrogen oxidation reaction (HOR) catalysis in alkaline electrolytes. The authors reported that the two nanoalloy catalysts showed very high intrinsic activities towards HOR activity compared to most of previously reported non-previous metal catalysts. The experimental results are important and sufficient characterizations, tests and computations were provided. It can be published in Nature Communications after considering the following questions.

Response: We greatly appreciate the reviewer's high praise and support on the publication of this work.

1) The authors should decouple the effects of composition, structure and morphology. Since a microwave synthesis method was used to prepare high surface area of MoNi_4 and WNi_4 samples, What would the HOR activity be if they are just non-porous nanoparticles (not the nanosheet structure)? A blank experiment on HOR activity test on $\text{MoNi}_4/\text{WNi}_4$ nanoparticles should be given. In addition, what is the composition effect? It seems that the authors have optimized the Mo-Ni composition, however, what is the reason behind the optimal composition? Is it because of change of the ratio of surface Ni, of the phase structure, or of chemical state of Ni, or some other reasons?

Response: We appreciate the reviewer for the thoughtful comments and questions. To synthesize the desired nanoalloys, we first prepared the sheet-like Mo(W)-doped $\text{Ni}(\text{OH})_2$ precursors, which were then annealed in H_2/Ar atmosphere to generate alloys. The high-temperature treatment leads to MoNi_4 porous sheets (composed of nanoparticles actually; see **Supplementary Fig. 3**) and WNi_4 nanoparticles (**Supplementary Fig. 4**). Following your suggestion, we further prepared non-porous MoNi_4 nanoparticles (refer to *ACS Catal.* **2013**, 3, 166-169; see **Figure R11** below) and compared their HOR activity with that of MoNi_4 porous sheets. We see that the non-porous MoNi_4 nanoparticles also bear certain HOR activity in alkaline electrolytes but the reactivity is inferior to our designed alloyed catalyst. These results suggest that the MoNi_4 phase is indeed HOR active, and

the porous sheets composed of nanoparticles could provide much more active sites, thus giving superior HOR performances.

Figure R11. Physical characterization and HOR activities of non-porous MoNi_4 nanoparticles. **a**, SEM image. **b**, XRD pattern. The inserted lines are indexed to MoNi_4 (JCPDS 65-5480). **c**, HOR polarization curve.

In our original manuscript, we performed a series of control experiments that determine the MoNi_4 and WNi_4 phases are optimum for HOR catalysis in alkali (see **Supplementary Fig. 22** and **23**).

Deviating this composition leads to inferior HOR activity. Our UPS and CO-stripping experiments, in conjunction with DFT calculations, uncover that MoNi₄ and WNi₄ structures simultaneously enable the optimized HBE and OHBE on their surfaces, which can substantially boost the Volmer step, leading to the marked HOR performance of the MoNi₄ and WNi₄ catalysts. Therefore, our studies indicate that altering the ration of Ni, oxidation state of Ni, or the phase structure will cause decreased HOR properties.

We have added the new data in the **revised SI (Supplementary Figure 26)** and provided some details over there.

2) *The authors provided the HOR durability tests, and showed both catalysts are better than Pt/C, what is the main mechanism behind it? the authors should show some characterizations after HOR tests.*

Response: We thank the reviewer for the useful question and suggestion.

Well, in our original manuscript, we have provided SEM, TEM, HRTEM and STEM-EDX analyses of the MoNi₄ and WNi₄ alloys after HOR tests (see **Supplementary Fig. 30 and 32**), which show that the morphology, composition and structure of the alloyed catalysts are well maintained. We here further collected the cycled alloyed catalysts and carried out XRD characterizations (**Figure R12** below). Our results show that their crystal phases are also kept well after testing. These results clearly demonstrate the structural robustness of our alloyed catalysts that catalyze HOR in alkaline electrolytes.

Figure R12. XRD patterns. **a**, MoNi₄ after HOR test. The inserted lines are indexed to MoNi₄ (JCPDS 65-5480). **b**, WNi₄ after HOR test. The inserted lines are indexed to WNi₄ (JCPDS 65-2673).

By contrast, we observed that commercial Pt/C catalyst shows gradual current drop during the stability assessment. To probe the intrinsic reason, we performed post-TEM analysis of the cycled Pt/C catalysts. As shown in **Figure R13**, we find clear particle size agglomeration as reaction time prolongs. After reacting from 0 h to 20 h, the Pt particle size grows obviously and forms agglomerated clusters, which could be the intrinsic reason that leads to inferior stability of the Pt/C catalyst.

Figure R13. TEM images for commercial Pt/C after 0 h (a), 10 h (b) and 20 h (c) of stability test.

We have added these new data in the **revised SI (Supplementary Figure 28, 31)** and provided some discussion properly over there.

3) *On the DFT computation, it is convincing that both HBE and OHBE play roles on HOR in alkaline electrolyte. Can the authors used DFT to give some insights into the CO-tolerant capability of the Mo(W)-Ni nanoalloy catalysts?*

Response: We thank the reviewer for the positive comment and good suggestion. Following your suggestion, we performed DFT calculations on the studied catalysts to probe the CO adsorption ability, which could offer insights on their CO-tolerant capability. We created the models with adsorption of OH intermediates to represent the catalysts in alkaline environment (see **Figure R14a-c**). Then, CO molecules were absorbed on the surfaces of these catalysts to form CO* intermediates (see **Supplementary Fig. 33** for details). Our calculated adsorption energies for CO* intermediates are -1.68 eV, -1.45 eV and -1.44 eV for Pt(111), MoNi₄(211), and WNi₄(211) surfaces (**Figure R14d**), respectively. These results show that CO* adsorbed on Pt surface is too strong, which covers the Pt active sites, leading to HOR deactivation. By contrast, CO adsorption on MoNi₄ and WNi₄ catalysts is much more appropriate, showing the CO-tolerant capability.

Figure R14. Model and calculated results of CO adsorption on catalysts. **a**, Pt(111). **b**, MoNi₄(211). **c**, WNi₄(211). **d**, Calculated CO adsorption energies on catalysts.

We have added these new data in the **revised SI (Supplementary Figure 33)** and provided some discussion over there.

4) The authors showed the geometric activity of HOR on MoNi₄ is greater than a commercial Pt/C catalyst, however, they did not show the true intrinsic activity comparison between MoNi₄ and Pt/C. *Supplementary Figure.16* should include the HOR activity on state-of-the-art Pt/C catalysts. Make sure that all the HOR activities on various non-previous metal catalysts were tested under same conditions.

Response: We thank the reviewer for the useful suggestion. We accepted your suggestion and added the intrinsic activity of Pt/C catalyst for comparison (**Figure R15**). We obtained the electrochemical

active surface area of Pt/C by a commonly used CO-stripping route (see **Figure 5b** in the MS), assuming a specific charge of $420 \mu\text{C cm}^{-2}_{\text{Pt}}$ for stripping CO monolayer on Pt. **Figure R15** shows the ECSA-normalized exchange current density of various catalysts that gained at the same condition. We find that the ECSA-normalized exchange current densities are 0.065 and 0.068 mA cm^{-2} for MoNi₄ and WNi₄ catalysts, which are larger than that of 0.043 mA cm^{-2} for Pt/C catalyst, suggesting the superior intrinsic HOR activity of our designed alloyed catalysts.

Figure R15. Comparison of the intrinsic HOR activities—ECSA normalized exchange current density—of MoNi₄, WNi₄, freshly-synthesized Ni, Pt/C and most reported PGM-free HOR catalysts measured in alkaline electrolyte (pH 13).

This new Figure was updated as **Supplementary Fig. 19** in our **revised SI**.

5) *It will be more exciting to see the new bimetallic catalysts tested in a realistic fuel cell, and show they can serve as a promising non-precious anode for realistic AEMFCs.*

Response: We thank the reviewer for the nice suggestion. Following your suggestion, we collaborated with colleagues and carried out the AEMFC tests. The AEMFC was achieved by using the quaternary ammonia poly(Nmethyl-piperidine-co-p-terphenyl) membrane and ionomer which need be transferred from Cl⁻ to OH⁻ conductivity in KOH electrolytes before use. As shown in **Figure R16**, the cells show maximum power densities of 45 mW cm^{-2} for MoNi₄ anode and 37 mW cm^{-2} for WNi₄ anode with a high open circuit voltage (OCV) of above 1.0 V, whereas the cell with the Ni anode shows an extremely low OCV of 0.2 V and gives no current curves. Despite the AEMFC performance need be further improved by optimizing numerous parameters, yet we can see that the alloyed catalysts performed obviously superior to single Ni catalyst.

Currently, the COVID-19 pandemic has led to the laboratory shutdown, we are unable to systematically modify the experimental parameters for these new PGM-free catalysts although much

improvement is expected. As a matter of fact, optimizing the MoNi₄ and WNi₄ catalysts and exploring their use in AEMFCs would be our future works, which need comprehensive and systematic research and we hope to report related results later separately. We thus have not included this very preliminary results in the present work.

Figure R16. Polarization curve (no *iR*-corrections) of the AEMFC with MoNi₄ and WNi₄ anodes, respectively. The alloy loading in anode is 2 mg cm⁻², whereas the Pt loading in cathode is 0.4 mg cm⁻². The back pressure of gases is 1 atm at both sides of the cell. The operating temperature is 80 °C. Pure H₂ and O₂ are fully humidified and fed into the cell at a flow rate of 500 mL min⁻¹.

Reviewer #3 (Remarks to the Author):

*This paper demonstrated that the nickel-molybdenum nanoalloy with tetragonal MoNi₄ phase can catalyze the HOR efficiently in alkaline electrolytes. This catalyst is tolerant to carbon monoxide poisoning. DFT calculations were performed to evaluate the HBE and OHBE. The authors rational the good efficiency in the combination of nickel and molybdenum for optimized adsorption of intermediates. Another analysis tool used is the *d*-band center shift trend. Some points for minor improvement:*

Response: We greatly appreciate the reviewer for the positive feedbacks on the contents presented in our manuscript.

1. Please clarify why no XC corrections were included for transition metals.

Response: We thank the reviewer for the thoughtful suggestion.

In order to ensure the correctness of our calculations and the calculation results are comparable, the XC correction treated for the transitional metals was based on the previous literatures (*Nat. Commun.* **2017**, 8, 15437; *Energy Environ. Sci.*, **2014**, 7, 1719-1724; *Nat. Commun.* **2016**, 7, 10141). These works includes the same transition metals as ours, for instance, Mo, Ni, Pt, MoNi and CoNi alloys in *Energy Environ. Sci.*, **2014**, 7, 1719-1724; MoNi, Ni, Mo in the *Nat. Commun.* **2017**, 8, 15437. In addition, the hydrogen adsorption energies of Ni(111), Mo(110) are -0.57eV and -0.72eV, respectively, in the work of *Nat. Commun.* **2017**, 8, 15437, and those of Ni(111), Mo(110) and Pt(111) are -0.51eV, -0.70eV and -0.46eV in *Energy Environ. Sci.*, **2014**, 7, 1719. Our calculated hydrogen adsorption energies are close to their results (-0.56 eV for Ni(111), -0.74 eV for Mo(110) and -0.47 eV for Pt(111)), confirming the reliability of our results.

We have added the explanations in the revised manuscript as following “The exchange correction treated for the transitional metals was based on the previous literatures which includes the same metals as ours for reliability and comparability.” (see the **revised MS, Pages 21**).

2. Please comment on the choice of 0.02 eV/Å for forces. This may be low.

Response: Thanks for the reviewer’s insightful and careful comments. The convergence criterion on forces for the calculations of the metal surfaces were chosen according to the previous literatures (*Nat. Commun.* **2017**, 8, 15437 (0.02 eV/Å); *Nat. Commun.* **2016**, 7, 10141 (0.05 eV/Å); *Energy Environ. Sci.*, **2015**, 8, 177 (0.02 eV/Å)). Therein, they have studied the same transition metal surfaces as ours, for instance, Mo(110), Ni(111) in *Nat. Commun.* **2017**, 8, 15437; Pt(111) in *Energy Environ. Sci.*, **2015**, 8, 177.

Following the reviewer’s suggestion, we also calculated the energies of metal surface with much higher convergence criterion. Taking Pt(111) as an example, the energies of Pt(111) with four layers are -207.22345487 eV, -207.22358964 eV and -207.22368742 eV with the convergence criterion of 0.02 eV/Å, 0.01 eV/Å and 0.005 eV/Å, respectively. We find that these energies are mere slightly different, implying the convergence of our calculation results. Thus, considering the computation cost and efficiency, we chose 0.02 eV/ Å as the convergence criterion on forces. We have added “These convergence criteria were chosen according to the previous literatures to ensure the accuracy” in **Pages 21** of the **revised MS**.

3. It is not clear to me which reaction mechanism was considering for modeling HOR. The analysis includes only the OH and H adsorption steps.

Response: Thanks for the reviewer’s insightful comments.

Commonly, HOR is thought to proceed via two reaction mechanism: Tafel–Volmer mechanism or Heyrovsky–Volmer mechanism (*Sci. Adv.* **2016**, 2, e1501602). According to previous work (*Adv. Mater.* **2019**, 31, 1808066), one can probe the reaction mechanism from the symmetry of Tafel plots and Tafel slopes of the catalysts. We find that the Tafel plot of MoNi₄ shows asymmetric $|j_k| \sim E$ relation, and its Tafel slope is $\sim 48.6 \text{ mV dec}^{-1}$, reasonably close to 39 mV dec^{-1} (**Figure R17**), suggesting that the HOR on the MoNi₄ catalyst is conducted via a Heyrovsky–Volmer pathway with a Volmer rate-determining step (*Adv. Mater.* **2019**, 31, 1808066). On the basis of above results, we conclude that the Heyrovsky–Volmer mechanism works for alkaline HOR on our alloy catalysts.

Figure R17. HOR/HER Tafel plot of the kinetic current density on MoNi₄ in H₂-saturated 0.1 M KOH.

Note:

The Tafel slope is obtained in the overpotential region from 50 mV to 100 mV to avoid the incorrect fitting in too low overpotential region based on the definition of Tafel equation (*J. Electrochem. Soc.* **2010**, 157, B1529-B1536).

Reviewers' Comments:

Reviewer #2:

Remarks to the Author:

The reviewer has no more questions, and supports its publication in Nature Communications.

P. S. The point-to-point answers to the referees' comments

REVIEWERS' COMMENTS:

Reviewer #2 (Remarks to the Author):

The reviewer has no more questions, and supports its publication in Nature Communications.

Response: We thank the reviewer for strong support on the publication of this work.